# Perlite and Rice Husk Ash Re-Use As Fine Aggregates in Lightweight Aggregate Structural Concrete—Durability Assessment

Maria C. Stratoura [1,2], Gerasimina-Ersi D. Lazari [2], Efstratios G. Badogiannis [2,*] and Vagelis G. Papadakis [1,3]

[1] Department of Environmental Engineering, University of Patras, GR30100 Patras, Greece
[2] School of Civil Engineering, National Technical University of Athens, GR15773 Athens, Greece
[3] Department of Civil Engineering, University of Patras, GR26500 Patras, Greece
* Correspondence: badstrat@central.ntua.gr; Tel.: +30-210-772-1266

**Abstract:** In this paper, perlite mining and rice production by-products, namely run-of-mine perlite and rice husk ash, are used as fine aggregates in combination with pumice and calcareous aggregates to produce lightweight concrete. Their use is evaluated mainly in terms of the durability of the concrete, by comparing four optimized lightweight concrete mixtures of similar density and strength with a reference one of normal weight. The sorptivity due to capillary sorption, open porosity, chloride migration, penetration resistance, and freeze and thaw response were studied to evaluate the durability of the lightweight concrete. According to the experimental results, the examined mixtures developed an adequate strength in order to be classified into strength classes greater than LC25/28 and, therefore, be used in structural applications. The durability of the mixtures was also sufficient, especially as far as the chlorides' penetration resistance is concerned, which was found to be up to 39% lower compared to the reference mixture. The sorptivity and open porosity of the LWC mixtures increased due to the porous nature of the lightweight aggregates, and the mixtures were also found to be susceptible to freeze and thaw cycles. Exceptionally, the lightweight concrete mixtures comprising pumice and perlite exhibited a lower sorptivity and resistance to chloride penetration than the standard concrete and a promising tolerance to freezing and thawing. Thus, the optimized combination of pumice and perlite is a sustainable recommendation for structural lightweight concrete production and use, promoting the wider exploitation of natural aggregates with an acceptable compromise on strength and durability.

**Keywords:** lightweight concrete; sustainability; pumice; perlite; rice husk ash; durability





## 1. Introduction

The impact on the environment stemming from the construction and operation of buildings is significant, accounting for 1/3 of greenhouse gas emissions and for 40% of the global energy consumption [1]. Parliaments have adopted specific structural codes [2], taking into account the technical and functional requirements of a building. The Zero Energy Building (ZEB) concept, a model building established by Belussi et al., 2019 [3] in which the functional needs and self-sufficiency of a building under certain functional conditions are ideally balanced, could be used as a reference in order to reach the objectives for energy efficiency in buildings.

It is, therefore, necessary that buildings must be designed to have a minimal impact on the natural environment [4]. The sustainable design of structures can substantially decrease the life-cycle, economic and environmental losses [5], and enhance the energy efficiency of buildings by using high-performance, durable, sustainable and thermally insulating construction materials [6].

In 2018, approximately 0.25 billion $m^3$ of sand was used in the production of concrete [7], one of the major sources of raw material use [8]. Despite the efforts of the industrial

sector globally to ameliorate its manufacturing processes, the need for raw materials is expected to double by 2060 [9]. It is, therefore, paramount to reduce the consumption of primary raw materials, propose the use of alternatives, such as lightweight aggregates from recycled materials, and, therefore, contribute to a reduction in our carbon footprint and fight climate change.

Lightweight concrete (LWC) is a type of concrete that can offer both technological and environmental benefits. Specifically, structural LWC offers flexibility in design and is cost effective due to the decreased inert load, enhanced seismic behaviour, wider openings, improved fire resistance, finer sections, lower storey height, lighter structural members, fewer steel reinforcements, superior durability and lower foundation costs [10,11]. Since the 1950s, structural LWC has been successfully introduced into the concrete industry for high-rise frames and floors, long-span bridges, off-shore infrastructures and prestressed or precast modules of all types [12]. However, making use of industrial by-products as aggregates for the production of LWC (e.g., rice husk ash and run-of-mine perlite), could reduce the environmental impact of the concrete industry, proving LWC to be a highly sustainable material.

Rice husk ash (R) is an agro-industrial residue deriving from the incineration of rice husk, and since it comprises 1/4 of the weight of rice, for every 1000 kg of husk burnt, approximately 250 kg of R is manufactured [13]. Roughly 35 million tonnes of R per year accumulates globally from the production of rice alone [14]. Specifically, in Greece, approximately 240,000 tonnes of rice is produced annually and there is a potential annual accumulation of 12,000 tonnes of R [15]. The bulk density of R is very low, while its content of silica is high [16]. It can, therefore, provide supplementary CSH and other pozzolanic products for the concrete matrix [17,18]; together with the massive quantities of R found globally, this makes it a promising and highly environmentally friendly material to be used as an aggregate in the production of concrete. However, there is a limited number of studies in the literature [19,20] that explore the potential use of coarser R as a fine aggregate in LWC and so additional research is required.

The fine aggregates in concrete could also be substituted by run-of-mine perlite (Pe), a raw perlite stream with a size of 0–2 mm or 0–4 mm and discarded as waste prior to the expansion process of perlite. The rejected amounts depend on the design of each mine's production line and can reach up to 10% by weight of the feeding perlite. In Greece, the annual perlite production capacity approaches approximately 700,000 tonnes [21], and considering an average rejection level of 5%, approximately 35,000 tonnes of run-of-mine perlite are rejected annually, which astonishingly remain unused. A few studies have focused on the use of ultra-fine expanded [22] or raw [23] perlite as an effective supplementary cementitious material, and have shown potential. Additionally, waste-expanded perlite has been studied as a replacement for quartz sand in the production of autoclaved aerated concrete [24] and as a partial or complete substitution for feldspar in the production of ceramic tiles [25]. Preliminary studies on using finer-than-4 mm perlite as a fine aggregate in LWC have shown positive results in terms of durability and strength [20]; however, further research is necessary.

Pumice, a natural volcanic material derived from the release of gas during the solidification of lava, has also been used in many countries around the world as an aggregate of LWC [26]. Greece has a notable pumice mining capacity, with a capacity of 1 Mt in 2020 [21], yet the local production of pumice LWC is rather limited. Pumice usage as either an aggregate or a mineral addition to concrete is a sustainable solution for the production of durable concrete [27]. Other studies have also demonstrated that the addition of ground pumice improves the mechanical properties and durability of LWC [28,29].

The present study examines how the combination of pumice with Industrial by-products, such as perlite mining fines and rice husk ash, could deliver durable LWC mixtures for structural use. The suggested by-products, as well as the LWC, are both materials of current interest and their proposed exploitation promotes a circular economy, as the produced concrete could be considered a material of increased sustainability.

## 2. Materials' Mix Design and Experimental Methods

### 2.1. Materials

Ordinary Portland cement (CEM I 42.5R) denoted as C, was added in all mixtures. Three different types of lightweight aggregates (LWAs) were examined in this study: pumice (Pu), Run-of Mine-perlite (Pe) and rice husk ash (R) with a (nominal) grading of 0–8 mm and 0–16 mm, 0–2 mm and 0–4 mm, and 0–2 mm, respectively. Three locally available crushed calcareous limestone aggregates (Ca) of 0–4, 0–8 and 0–16 mm gradings were also used. Photos of the LWAs are shown in Figure 1, Table 1 presents the chemical composition of the aggregates and the cement used in this study, while their physical properties (apparent density, $\rho_a$ and water absorption) are presented in Table 2. Finally, commercial super plasticizer (SP) and plasticizer (PL) were also used.

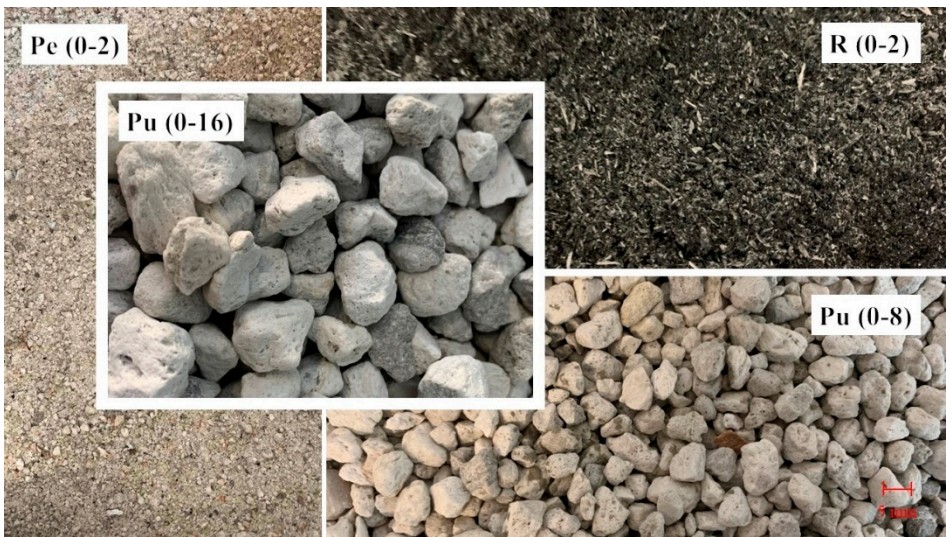

**Figure 1.** Lightweight aggregates.

**Table 1.** Chemical analysis (% *w/w*) of the concrete constituents.

| Constituent | SiO$_2$ | Al$_2$O$_3$ | Fe$_2$O$_3$ | CaO | MgO | SO$_3$ | K$_2$O | Na$_2$O | TiO$_2$ | P$_2$O$_5$ | L.O.I. |
|---|---|---|---|---|---|---|---|---|---|---|---|
| C | 18.95 | 5.22 | 3.27 | 62.51 | 2.20 | 3.63 | 0.51 | 0.38 | 0.34 | 0.09 | 2.90 |
| Pu | 68.21 | 11.83 | 1.15 | 4.09 | 0.44 | - | 4.00 | 2.76 | 0.12 | 0.03 | 7.37 |
| Pe | 73.81 | 12.97 | 1.00 | 1.40 | 0.25 | - | 3.49 | 4.49 | - | - | 2.59 |
| R | 90.61 | 0.08 | 0.45 | 1.22 | 0.48 | - | 0.84 | 0.66 | - | - | 5.66 |

**Table 2.** Physical properties of aggregates.

| Properties | Aggregates | | | | | | | |
|---|---|---|---|---|---|---|---|---|
| | Pu (0–8) | Pu (0–16) | Pe (0–2) | Pe (0–4) | R (0–2) | Ca (0–4) | Ca (0–8) | Ca (0–16) |
| **Density, $\rho_a$ (t/m$^3$)** | 1.61 | 1.13 | 2.20 | 2.15 | 1.79 | 2.67 | 2.67 | 2.61 |
| **Water absorption (%)** | 19.2 | 17.0 | 4.0 | 3.4 | 13.7 | 2.1 | 0.9 | 0.9 |

### 2.2. Mix Design

For the purposes of this study, four LWC mixtures (PeCaPu, PuCa, PePu, RPePu) containing different types of LWAs were compared to a concrete mixture of normal weight with limestone aggregates (REF). A batch of 25 l was produced for each composition and tested according to the experimental program described in the following section. Table 3 presents the proportions of each concrete mixture, designed according to the previous experimental work of Stratoura et al., 2018 [29]. The produced mixtures have a similar density (<D1.6) and

are expected to develop a sufficient compressive strength to be classified in a strength class above LC25/28, as specified in EN206-1 [30]. The Fuller theory was followed for the aggregate blending, delivering continuous curves of aggregate blends, as shown in Figure 2, and producing concrete mixtures with densities similar to the target density class. It should be mentioned that there are differences in the crushing indices of the produced LWC mixtures. Similar mixtures were produced in a preliminary study [20], albeit under different mixing conditions, and developed similar compressive strengths and highly comparable results, indicating that the effect of the different crushing indices is negligible.

**Table 3.** Syntheses of the designed concrete mixtures (kg/m$^3$).

| Constituent (Kg/m$^3$) | Mixtures | | | | |
| --- | --- | --- | --- | --- | --- |
| | REF | PeCaPu | PuCa | PePu | RPePu |
| C | 400 | 400 | 400 | 400 | 400 |
| Ca (0–4) | 946 | - | - | - | - |
| Ca (0–8) | 568 | 55 | 51 | - | - |
| Ca (0–16) | 370 | - | - | - | - |
| Pu (0–8) | | | 607 | 397 | 402 |
| Pu (0–16) | - | 492 | 354 | 446 | 402 |
| Pe (0–2) | - | 109 | - | 149 | - |
| Pe (0–4) | - | 437 | - | - | 101 |
| R | - | - | - | - | 101 |
| PL | 1.2 | 1.2 | 1.2 | 1.2 | 1.2 |
| SP | 4.8 | 3.0 | 2.4 | 2.4 | 3.2 |
| prW | 29 | 98 | 177 | 158 | 162 |
| Effective water | 160 | 160 | 160 | 160 | 160 |
| Total water | 189 | 258 | 337 | 318 | 322 |
| w/c (effective) | 0.4 | 0.4 | 0.4 | 0.4 | 0.4 |

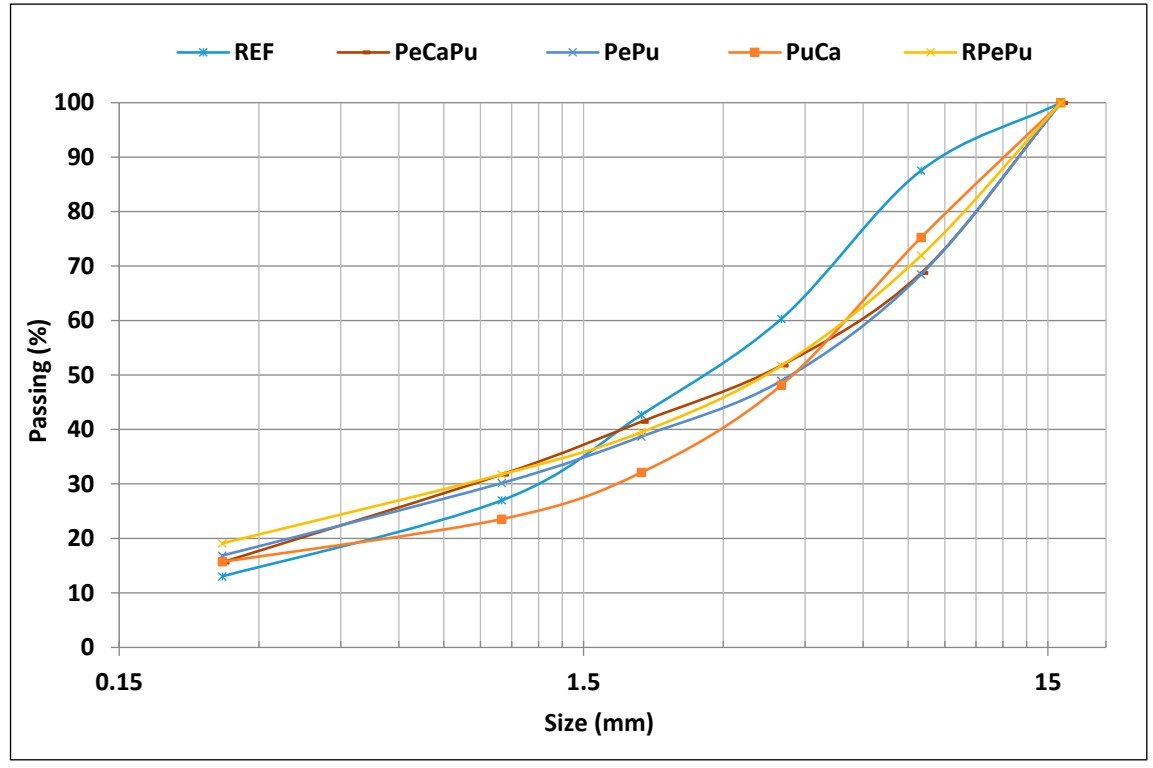

**Figure 2.** Grain size distribution of the aggregate blends used in each LWC mixture.

In all mixtures, both the cement content and the effective w/c were kept constant, considering the significant absorption potential of the LWAs (Table 2) that affects both the rheology of fresh concrete and the effective water for cement hydration. Specifically, the absorption capacity of the aggregates was estimated, and the corresponding water was pre-mixed with the aggregates as a pre-conditioning water (prW), 20 min before the final mixing of each LWC mixture. Table 3 presents analytically the pre-conditioning and effective water amounts. The total water shown in Table 3 is the sum of all the added water. The rheology of all the LWC mixtures was controlled by using appropriate amounts of SP and PL.

### 2.3. Test Procedures

Table 4 summarizes the test parameters with respect to the measured properties and their units, the specimen types and sizes, the testing age and the adopted standards.

**Table 4.** Details on the test parameters.

| Properties | Unit | Sample Age (Days) | No of Samples (per Age) | Size of Specimen | Standard |
|---|---|---|---|---|---|
| Slump Test | mm | - | - | - | EN 12350-2 |
| Density | kg/m$^3$ | - | - | - | EN 12350-6 |
| Air Content | % | - | - | - | EN 12350-7 |
| Compressive Strength | MPa | 28 and 90 | 3 | $100 \times 100 \times 100$ mm | EN 12390-3 |
| Open Porosity | % | 28, 90 and 180 | 2 | $\varnothing 100 \times 30$ mm discs [1] | ASTM C642 |
| Sorptivity Coefficient | mm/min$^{0.5}$ | 28 | 2 | $\varnothing 100 \times 30$ mm discs[1] | ASTM C1585 |
| Chloride migration test | m$^2$/s | 28, 90 and 180 | 2 | $\varnothing 100 \times 50$ mm discs [1] | NT Build 492 |
| Chloride ion penetration test | Coulomb | 90 | 3 | $\varnothing 100 \times 50$ mm discs [1] | ASTM C1202 |
| Freeze and Thaw test | - | >90 | 3 | $70 \times 70 \times 70$ mm | ASTM C666 |

[1] $\varnothing 100 \times 50$ mm or 30 mm discs were cut from the $\varnothing 100 \times 200$ mm cylindrical specimens.

#### 2.3.1. Fresh Concrete and Strength Tests

The workability of the fresh concrete was evaluated by using the slump test, according to EN 12350-2 [31]. Fresh density and air content tests were also carried out following the specifications of EN 12350-6 [32] and EN 12350-7 [33], respectively. The compressive strength of the concrete was measured at 28 and 90 days, in accordance with EN 12390-3 [34]. Three specimens per batch were tested at each age and the average of the three values is reported hereinafter.

#### 2.3.2. Open Porosity and Sorptivity Coefficient

The open porosity (OP) was evaluated according to ASTM C642 [35], by weighting in air and in water, and using a fully saturated $\varnothing 100 \times 30$ mm disc. The specimen was oven dried at 105 °C until a constant mass was achieved, whereas the duration of drying varied between 4 and 7 days. The OP(%) was calculated using the three weighted masses of two specimens ($\varnothing 100 \times 30$ mm discs) from each mixture at the age of 28, 90 and 180 days. The sorptivity, a fundamental parameter in controlling liquid transport in concrete and thus its durability, was measured according to ASTM C1585 [36]. The test was performed on similar dried concrete specimens ($\varnothing 100 \times 30$ mm discs, two from each mixture) after their dry mass stabilized. The specimens were peripherally sealed and rested on rods to allow free access to no more than 3 mm height tap water, only from one circular surface. The change of each specimen's weight due to the uniaxial water uptake (capillary absorption was recorded to calculate the sorptivity, S (mm/min$^{0.5}$). Specimens from each mixture were tested at the age of 28 days.

#### 2.3.3. Chloride Migration and Penetration Tests

The chloride penetration coefficient, $D_{nssm} (\times 10^{-12} m^2/s)$ was measured by performing a non-steady-state Rapid Chloride Migration Test (RCMT), in accordance with the NT Build 492 method [37]. A segment of 50 mm was extracted from the middle zone of a cylindrical

specimen ($\varnothing 100 \times 200$ mm) and was sustained for 24 h under a stable current difference, between a cathode solution of 10% sodium chloride (NaCl) by mass and an anode solution of sodium hydroxide (NaOH), 0.3 N. A total of six specimens were tested for each mixture, two for each test age of 28, 90 and 180 days. The specimens were kept in a calcium hydroxide solution until they were tested. After the completion of each test, the chloride migration depth was determined using the colorimetric method. Specifically, the specimen was split axially into two pieces and one of the two fractured surfaces was sprayed with a 0.1 M silver nitrate ($AgNO_3$) solution. The average depth of chloride penetration was determined from the change in colour in the area where the presence of chlorides chemically led to the formation of white silver chloride (AgCl). The classification of the chloride ion ingress resistance according to the aforementioned, was proposed by Nilsson et al. [38], as presented in Table 5.

**Table 5.** Classification of the Chloride ion ingress resistance for RCMT [37] (left) and RCPT [39] (right).

| NT Build 492 | | | ASTM C1202 | | |
|---|---|---|---|---|---|
| **Chloride Migration Coefficient** ($\mathbf{D_{nssm}}$)$\times\mathbf{10^{-12}}$ | **Class** | **Description** | **Charge Passed (C)** | **Class** | **Description** |
| >15 | L : | Low | >4000 | H : | High |
| 10–15 | M : | Moderate | 2000–4000 | M : | Moderate |
| 5–10 | H : | High | 1000–2000 | L : | Low |
| 2.5–5 | VH : | Very High | 100–1000 | VL : | Very Low |
| <2.5 | EH : | Extremely High | <100 | N : | Negligible |

The electrical indication of the concrete's capability to resist chloride ion penetration, measured in Coulombs (C), was also evaluated by carrying out a Rapid Chlorides Penetration Test (RCPT). As described in ASTM C1202 [39], concrete cylinders of 100 mm in diameter and 50 mm in height from each mixture at the age of 90 days were placed in a hermetically sealed cell, which contained a sodium chloride solution (3.0% NaCl) at the negative pole, and a sodium hydroxide solution (0.3 N NaOH) at the positive pole. A current of 60 V DC was applied and a charge flow passed from the negative to the positive pole through the specimen. A computer software reported the electrical charge that passed in 6 h. The resulting charge values were used for the classification of the samples, according to the scale offered by the standard [39], also shown in Table 5.

2.3.4. Freeze and Thaw Test

For each mixture, three cubic specimens with dimensions of $70 \times 70 \times 70$ mm$^3$ were used for the evaluation of the resistance of concrete to rapid freezing and thawing, according to ASTM C666 [40]. The standard describes rapid freezing and thawing in water, and the minimum and maximum target temperatures were set to $-18 \pm 2\,°C$ and $+4 \pm 2\,°C$, respectively. Each cycle took 4 h to be completed. Every five cycles, the unit weight, length change and ultrasonic pulse velocity (UPV) were recorded, while the compressive strength, water absorption and sorptivity coefficient were measured after 75 cycles. The results of every measurement were compared to the control samples of each mixture, which were cured in water for the same duration. The average values of three samples were used for the evaluation of each property.

**3. Results and Discussion**

*3.1. Fresh Concrete Properties and Strength*

The properties of fresh and hardened concrete for all concrete mixtures are presented in Table 6. According to the slump test results, all the mixtures are classified above the S2 slump class. Noticeable results are reported for the mixtures PuCa and PePu, which exhibited a very high slump (S4), higher than normal concrete, even though less superplasticizer was added. Certainly, the pre-wetting process of the aggregates had a detrimental role

in both the enhancement of workability and the reduction in the slump settlement of the mixtures. The saturated aggregates prevented the mixtures from an initial or a delayed water absorption that would result in a looser slump. In addition, the higher percentages of air trapped in these mixtures improved the workability of the concrete [41].

**Table 6.** Test results of the optimized concrete mixtures in fresh and hardened state.

| Mixture | Slump (mm) | Air Content (%) | Density (kg/m$^3$) | Compressive Strength (MPa) | | | |
| | | | | $f_{c,28d}$ | stdev | $f_{c,90d}$ | stdev |
|---|---|---|---|---|---|---|---|
| REF | 110 | 3.1 | 2378 | 79.1 | 0.8 | 79.3 | 1.9 |
| PeCaPu | 170 | 4.8 | 1682 | 35.3 | 1.6 | 39.0 | 2.3 |
| PuCa | 240 | 6.2 | 1526 | 27.6 | 1.5 | 33.2 | 1.9 |
| PePu | 200 | 5.3 | 1527 | 26.0 | 0.2 | 32.6 | 2.4 |
| RPePu | 70 | 5.6 | 1553 | 26.8 | 1.6 | 30.3 | 2.3 |

A respective increase in the percentage of entrained air was noticed in the mixture RPePu, however, a slump settlement reduction of up to 48.5% was reported compared to the reference mixtures. Other studies have also reported similar observations, occurring from the replacement of LWAs with R in LWC mixtures [42]. This behaviour is attributed to the irregular shape of rice husk ash particles, as well as their highly microporous cellular structure, which both lead to an increase in the friction and water demand of the mixture [13].

As mentioned earlier, the values of air content in the LWC mixtures increased (4.8–6.2%) compared to the normal-weight concrete (3.1%). An increase in the air content is expected when LWAs are used in the concrete mixtures, since their porous structure traps air in the concrete matrix. Another fact explaining the increase in the air content is the addition of more superplasticizers, which are known for trapping air between the particles to reduce friction [42].

Finally, LWC mixtures achieved density values of 1526–1682 kg/m$^3$, which reduced up to 36% when compared to the REF mixture. All the mixtures satisfied the initial density goal of the mix design classified in the density classes below D1.6, according to EN206-1 [30].

The average compressive strength after 28 and 90 days ($f_{c,28d}$, $f_{c,90d}$) of curing for the LWC mixtures are presented in Table 6, and are compared to the compressive strength values of normal weight concrete. By blending aggregate mixtures with LWAs, the compressive strength of the LWC decreased, as a result of the weak nature of the LWAs. Specifically, at 28 days, the strength of the LWC ranged from 26.0 MPa to 35.3 MPa, which corresponds to less than half the strength of REF. PeCaPu could be, therefore, classified in the strength class LC30/33, whereas the mixtures PuCa, PePu and RPePu are classified in LC25/28. Therefore, all LWC mixtures could be used safely at various structural applications.

Among the LWC mixtures, PeCaPu exhibited the highest strength after 28 days (35.3 MPa). In terms of the strength of the mixtures, perlite satisfactorily replaced the calcareous sand in the aggregate blend. The PePu and RPePu mixtures developed an almost identical strength with PuCa, which confirms the use of Pe as an alternative fine aggregate material. Additionally, as shown on Table 6, the compressive strength of the LWC increased at the end of curing from 11 to 28%, which is significantly higher compared to the increase in REF (0.3%). The two mixtures with the highest percentages of pumice, PuCa and PePu, showed the greatest increase in compressive strength from 28 to 90 days. Particularly, the compressive strength of PuCa and PePu increased by 20% and 28%, respectively; and by increasing the volume of Pu and Pe in the aggregate blend, the strength increased further after curing. It is known from the literature that pumice significantly improves the compressive strength in the long term, due to its pozzolanic reactivity [20,43]. Moreover, the reactive silica content

of pumice provides better Si-O-Si bonding between the paste and the aggregates, as fresh cement paste penetrates into the open large pores of the LWA [44]. Additionally, the internal curing provided by the LWA has a positive effect on the development of the material's strength. Water released from the LWA hydrates the unhydrated cement particles, compensating for the strength development in the later hydration course [45]. On the other hand, the expected positive effect of R on the strength after the age of 28 days, which is reported in the literature to be in some cases even above 50% [46], was not observed in this study. Only a 12% strength increase in the 28-day compressive strength of RPePu was reported at the age of 90 days, which was lower in comparison with the other LWC mixtures, indicating low pozzolanic efficiency for the specific grading of R. The results are consistent with those of similar research [47] on normal weight concrete, where the reported strength increase from 28 and 90 days was close to 10%, when R replaces aggregates at a level of 20% by cement mass (here 25%). The same study [47] also presented the role of the fineness of R on its pozzolanic reactivity. Madani et al., 2012 [48] studied the effect of the coarseness of R when used as an aggregate. Coarse R is unable to produce a uniform distribution in the mixture of the fine particles, which act as a nucleation site for the pozzolanic reactions. The reported limited increase in the compressive strength stems from the reactive silica contained in R, as well as the internal curing provided by the reserved water in the porous skeleton of LWA [49].

### 3.2. Porosity and Sorptivity

The experimental results related to the open porosity measured different ages (28, 90 and 180 days) of concrete samples are presented in Figure 3, where the concrete mixtures are illustrated from left to right with descending density. The values obtained for the open porosity coefficient, after the 28th day, range from 20% to 34%, significantly higher compared to REF (13%). These results are in agreement with findings in the literature [20,29,50], indicating that the replacement of natural aggregates by LWA increases the water absorption of the LWC.

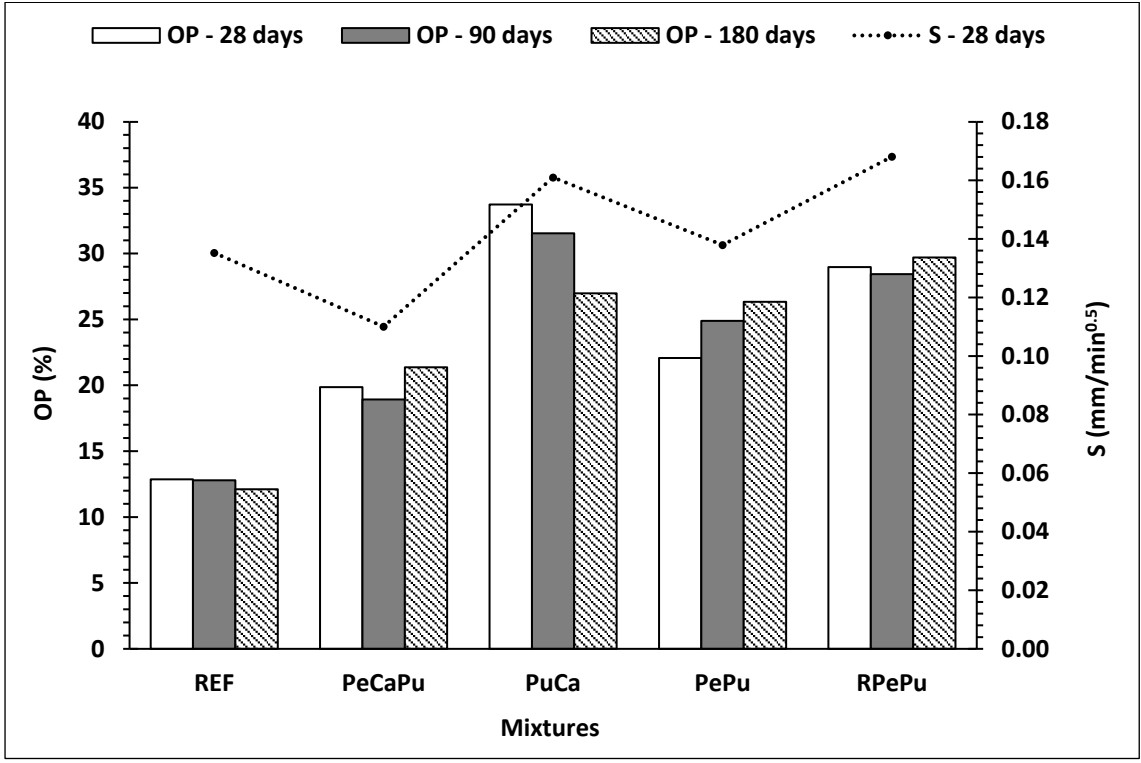

**Figure 3.** Effect of incorporating LWA on the open porosity and sorptivity at the ages of 28, 90 and 180 days, and 28 days, respectively.

Nevertheless, the combination of Pe (0–2) with Pu in the LWA blend reduced the open porosity, which also occurred in the concrete mixtures PeCaPu and PePu at all ages compared to the other LWC mixtures. As mentioned earlier, fine particles of Pe (0–2) functioned as filling particles for the paste's microstructure and provided a denser interfacial transition zone. Consequently, a reduced water absorption rate and lower open porosity values are measured.

In contrast, the observed increased porosity of the RPePu can be attributed to the highly porous structure of rice husk ash. Öz, 2012 [50], reported that unground R particles are largely irregular, vesicular, layered in appearance and exhibit a porous structure affecting negatively the concrete's homogeneity. Due to all the reasons mentioned, an increase (125%) in the RPePu's porosity is reported at the age of 28 days compared to conventional concrete.

Finally, in concrete specimens of older age (90 and 180 days), the measured values of open porosity seem to have improved slightly. For instance, the PuCa mixture shows a 20% reduction in the value of open porosity after 180 days. This mixture showed an improvement in the compressive strength in the long-term (see Section 3.1) as well.

In the same figure (Figure 3), the sorptivity, $S$ (mm/min$^{0.5}$), test results of the LWC specimens at the age of 28 days are presented. Throughout the test, the capillary water absorption of all the specimens increased and the coefficient of correlation ($R^2$) values were near 1, satisfying the high degree of linearity (0.98) that the standard [35] suggests. The obtained values for the coefficient of sorptivity ranged from 0.11 mm/min$^{0.5}$ to 0.17 mm/min$^{0.5}$. Between all the LWC mixtures, the sorptivity coefficient values varied with a very similar trend to that of the open porosity values, as both these phenomena are strongly related to the porous structure of concrete. A lower rate of capillary water absorption was reported for the PeCaPu mixture, even lower than that of conventional concrete (by 19%). Again, the fine grains of Pe (0–2) acted possibly as fillers and densified the interfacial zone.

At the same time, PuCa and RPePu mixtures exhibited higher sorptivity rates (0.16 mm/min$^{0.5}$ and 0.17 mm/min$^{0.5}$, respectively) than the conventional concrete (0.13 mm/min$^{0.5}$). Sorption increases with the increase in the amount of coarser pumice in the mixture, such as Pu (0–16) and Pu (0–8), because of its higher porosity. Interestingly, in the case of RPePu, the addition of Pe (0–2) did not exhibit the influence of the rice husk ash on the capillary pore system. These results are in agreement with those found in the literature, where a 2.5 times greater capillary water absorption value is reported when coarse pumice aggregates replace normal weight aggregates [51].

### 3.3. Chloride Penetration and Migration Tests

Chloride migration (RCMT) coefficients and penetration (RCPT) resistance were evaluated according to the relative standards, [37] and [39], respectively. Concrete mixtures were evaluated by the RCMT at the ages of 28, 90 and 180 days, while RCPT was conducted only at the age of 90 days.

The values obtained from the RCMT are presented in Figure 4. The chloride migration coefficient, $D_{nssm}$, values of the concrete samples at the age of 28 days ranged from 5.5 to 8.5 m$^2$/s; this was lower in most cases when compared to REF (9.0 m$^2$/s). The addition of Pu, Pe and R significantly improved the chloride resistance, evident even only after 28 days, although it increased further over time, contrary to the REF that did not show any significant improvement after 90 days of curing. The PeCaPu mixture had the best chloride penetration resistance response at all ages, with the lowest $D_{nssm}$ being 1.4 m$^2$/s measured at 180 days. PePu and PuCa showed the highest improvement (approx. 60%) of $D_{nssm}$ between 28 and 90 days, which is consistent with the improvements in strength and porosity of the same mixtures. Although the RPePu mixture did not show the highest chloride penetration resistance, and despite the fact it had the second highest $D_{nssm}$ after REF at the age of 28 days, at the age of 180 days, it demonstrated an impressive 74% increase in the $D_{nssm}$. The $D_{nssm}$ typically decreases with the exposure period independently of the type of aggregates used as a result of the pozzolanic reactivity, the binding capacity and the chloride pore-blocking effect of the LWA [52]. It is also well-established in the literature

that the time-dependent improvement in the permeability of LWC is attributed to the so called "internal curing effect", due to the delayed hydration of the cementitious materials caused by the available water in the LWA pores [45]. The internal curing effect and mostly the blocking effect of the LWA could also justify why, despite the higher open and capillary porosity of all the LWA concrete samples, and, therefore, their expected high penetrability, low values of $D_{nssm}$ are reported.

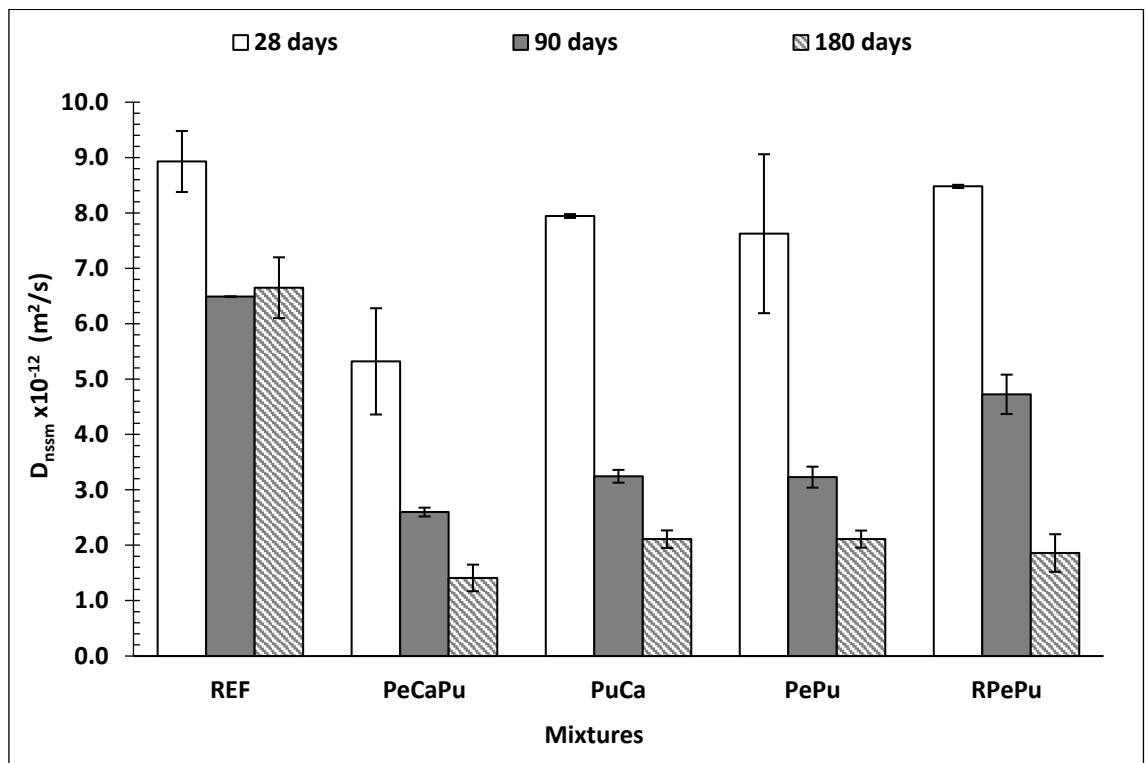

**Figure 4.** NT Build 492 chloride migration coefficient of the concrete mixtures at different curing ages.

The electrical charge measured by the RCPT for the concrete mixtures at 90 days is presented in Figure 5. These results are also directly compared to their counterparts obtained by the NT Build 492 [37] test, aiming to classify the concrete mixtures according to both classifications given by the ASTM C1202 standard [39] and Nilsson et al., 1998 [38]. As far as the chloride penetration resistance of the LWC mixtures is concerned, according to the ASTM C1202 standard [39], the electrical charge varied from 1525 to 1718 C. In comparison with the REF mixture, the incorporation of LWA reduced the penetrability of the chlorides into the LWC, by an average of 33.5%. Specifically, comparing the LWA mixtures, chloride penetrability was the lowest in RPePu (1597 C), while the highest one was recorded for PeCaPu (1718 C).

Table 5 shows a general correspondence between the results that can be obtained by the two different tests (RCMT and RCPT), which was also evident in the measurements of the chloride-ion penetrability on all LWCs at the age of 90 days; these were classified as of high (RCPT) to very high (RCMT) resistance.

Although the two methods differ in the specimen's initial condition (saturated and not saturated in $Ca(OH)_2$) and the testing parameters (solutions concentration, voltage and testing time), the results were found to be qualitatively similar. Conclusively, both methods certify that all the LWA mixtures exhibit excellent chloride penetration resistance after 90 days of curing, without any significant variation among the different mixtures.

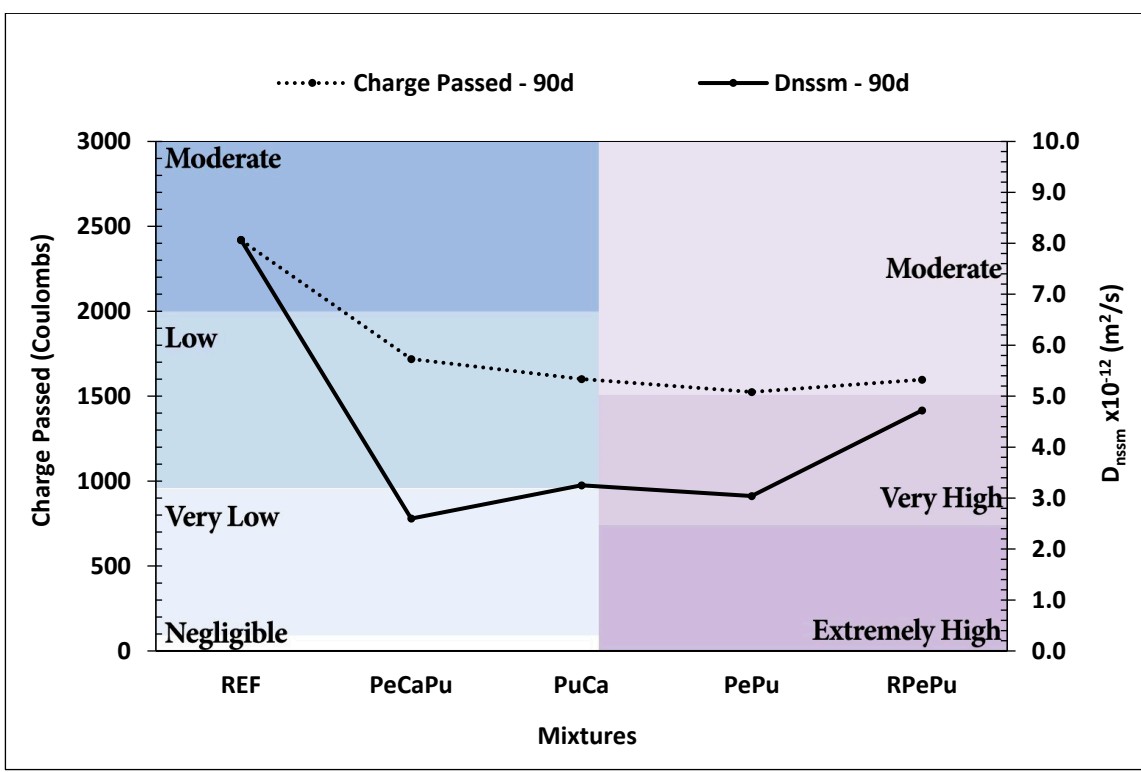

**Figure 5.** RCPT and RCMT results of the concrete mixtures at 90 days, according to the relative standards and their classification, as established by Nilsson et al. [38] and ASTM C1202 [39].

### 3.4. Freeze and Thaw Test

Ultrasonic pulse velocity (UPV) and freeze-thaw (FT) test results are presented and discussed in terms of total mass and total length change (% with respect to the initial mass and length of the specimen). In addition, the effect of 75 FT cycles on the open porosity (OP), sorptivity (S) and compressive strength ($f_c$) of the concrete is discussed.

#### 3.4.1. Mass and Length Change

The weight change in the LWC mixtures after the FT cycles is presented in Figure 6. In comparison to REF, the effect of the FT cycles on the specimens' mass was negligible apart from the RPePu mixture.

As shown in Figure 6, a negative mass change (%) occurred after the first FT cycles as expected, since the measurements are taken after the freeze cycle when the water in the specimens was frozen. The negative mass change (%) is consistent across all mixtures until the 75th cycle; this is except for the RPePu mixture, in which a positive mass change (%) was reported after the 10th cycle, which was kept increasing over time. After running 75 cycles of FT, the mass of the RPePu specimens increased by approximately 3.5%. It is clear that the addition of R has a negative impact on the response of the LWC specimens during repeated cycles of FT. The mass change of PeCaPu also reached a positive value once it completed the 75th cycle, a fact that signifies the initiation of the degradation of its structure.

Figure 7 presents the change in length (%) of the LWC mixtures during the FT test that was calculated by subtracting the length of the samples in a thawed condition from their initial one. All the mixtures, except RPePu, presented a similar trend in length change. The RPePu mixture showed an increase in the length over the limit of 0.1%, at the 65th cycle and onwards. According to the ASTM C666 [40], length changes greater than 0.1% reveal a structural degradation.

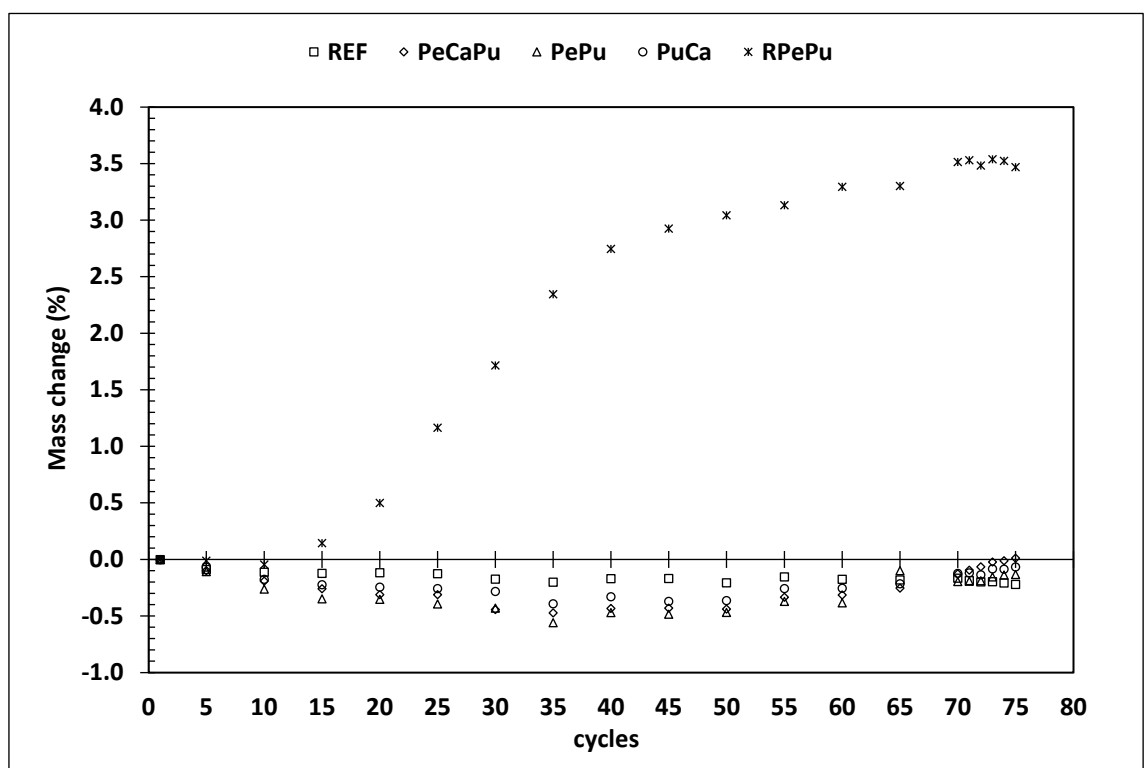

**Figure 6.** Mass change (%) of concrete specimens during freeze-thaw cycles.

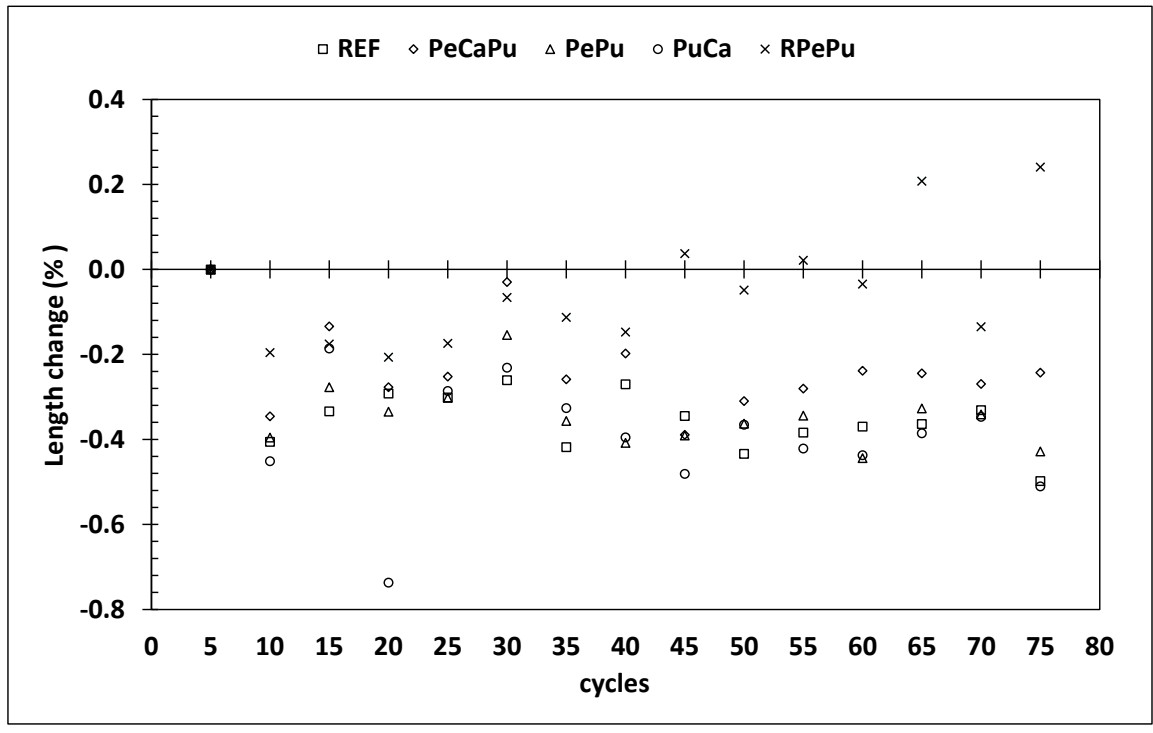

**Figure 7.** Length change (%) of concrete specimens during freeze-thaw cycles.

By observing the measured length change values across all mixtures, PeCaPu appears to have the largest resistance to the expansion effect of the FT cycles. In terms of mixture ingredients, PeCaPu contains a fine grade of perlite aggregate, Pe (0–2), which seems to have contributed positively to its endurance. In contrast, the results showed that by increasing the percentage of pumice in PePu and PuCa, the resistance of the LWC structures against

the FT cycles was not affected positively, a finding also highlighted in other studies [53]. LWAs demand large amounts of water during mixing and, therefore, absorb large quantities of water due to their porous nature. The water surplus contained in the pore system of the concrete promotes the FT degradation and, as the cement paste is susceptible to the water content, expansion is evident as a result of the ettringite and other hydrates [54]. However, until the 75th cycle, an insignificant length change (<0.1%) was reported in all concrete mixtures, except in RPePu.

### 3.4.2. Ultrasonic Pulse Velocity

The UPV (Km/s) values of concrete samples after each FT cycle are presented in Figure 8. The concrete mixtures showed UPV values inversely proportional to the densities of the mixtures. The REF mixture recorded the highest UPV value, whereas among the LWC mixtures, PeCaPu and RPePu registered the highest and lowest values, respectively. At the same time, the mixtures PuCa and PePu showed almost identical UPV values for most FT cycles.

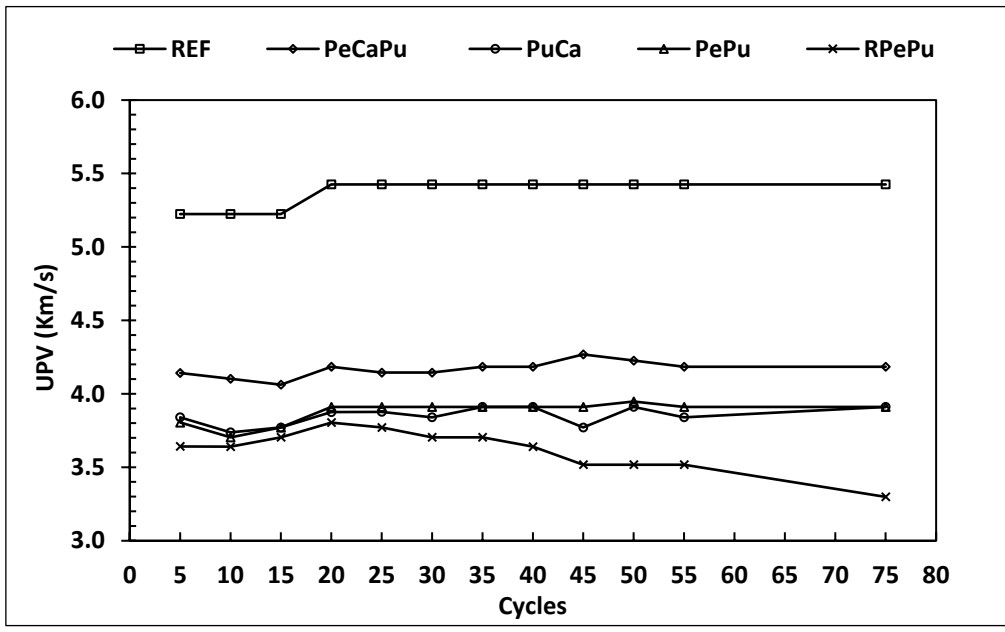

**Figure 8.** Ultrasonic pulse velocity (Km/s) of concrete specimens during freeze-thaw cycles.

Based on the experimental results, no significant change in the UPV values was recorded after running 75 FT cycles, and all mixtures exhibited a rather neutral trend of UPV change. The UPV values were expected to reduce, however, considering the structural degradation of the concrete skeleton due to the FT testing. It seems that the combination of pumice and perlite Pe (0–2) contributed positively to maintaining the quality of LWC mixtures during the FT test, although, according to the literature, the UPV values should decrease with increasing pumice replacement [55]. Only the RPePu mixture showed a constant reduction in the UPV values after the 20th cycle. Thus, the UPV values could not reveal the internal damage captured, however, by the compressive strength measurements, as will be discussed in the next Section 3.4.3.

It should be mentioned that the higher UPV values measured during the first FT cycles for most mixtures are attributed to frost formation, which is known to increase the UPV values of concrete [56].

### 3.4.3. Water Absorption and Compressive Strength

Changes occurring after the FT test with respect to the measured values at the beginning of the FT test for the compressive strength ($f_{c,FT}$), open porosity ($OP_{FT}$) and sorptivity ($S_{FT}$) are used as degradation indexes. Table 7 presents this change (%) for all concrete

mixtures. The strength is reported to be affected mainly after the FT test; therefore, $f_{c,FT}$, has a primary role in the ranking of concrete mixtures as far as their resistance against FT is concerned. The PePu mixture performed the best in three indexes, compared not only to the other LWC mixtures, but also to REF. Based on these findings, the combination of pumice and perlite could improve the resistance against FT degradation mechanisms.

**Table 7.** Degradation indexes (%) after freeze-thaw cycles of concrete mixtures.

| Mixtures | $f_{c,FT}$ | $OP_{FT}$ | $S_{FT}$ |
|---|---|---|---|
| REF | −9 | −1 | −46 |
| PeCaPu | −25 | 24 | 5 |
| PuCa | −19 | −9 | 15 |
| PePu | 7 | −26 | −8 |
| RPePu | −26 | −8 | 212 |

Since the compressive strength is a major index for the evaluation of the frost resistance of LWC, its variations are discussed further. The strengths of PuCa, PeCaPu and RPePu after 75 FT cycles dropped by 19%, 25% and 26%, respectively, compared to measurements taken at the beginning of the FT test for corresponding samples. According to the results, the combination of Pu, Pe and R decreased the resistance of LWC, indicating a reduction in durability.

As far as porosity is concerned, PeCaPu showed a change of +24% at the 75th cycle. In contrast, a maximum reduction (26%) of porosity was measured in the case of PePu at the end of the test, justifying the higher compressive strength that was recorded for the PePu mixture.

The same absorption behaviour was recorded for the PePu mixture during the sorptivity test, where the results showed an 8% reduction. As discussed above, the combination of Pu and Pe (0–2) demonstrates a greater effect on the LWC structure than the other LWAs due to the filler effect of Pe (0–2).

The sorption change (approx. +212%) of RPePu was the highest among the LWC mixtures, demonstrating the internal degradation of the specimen. According to standard [40], the experimental program should be terminated at the 65th cycle for the RPePu mixture. Consequently, the additional strain of the RPePu mixture during the last 10 FT cycles could be the cause of the high degradation of the capillary pores of the sample.

In general, the PePu mixture was the only one that exhibited endurance during the FT test. As discussed, the filler action of Pe (0–2) in the mixture led to a refinement of the pore system, a reduction in the permeability and a restriction of the entrapment of water that expands during the freezing cycle.

To conclude, all the LWC mixtures, excluding PePu, presented degradation after the FT test, corresponding to the adopted indexes. When the concrete is exposed to FT cycles, the ice forming in the capillary pores generates an expansive stress [57], which leads to the degradation of the concrete, as shown in the analysis of the results of the FT tests. Moreover, the FT cycles cause new cracks to form, which connect with existing ones, as well as the existing pore network of the concrete matrix. This results in an increase in the water absorption during thawing, which promotes the development and migration of frost [58]. The type of LWA can significantly influence the FT resistance of the LWC and the physical properties of the used LWA, with size and porosity being the most important factors.

## 4. Conclusions

In this study, LWC mixtures were designed to combine pumice and two industrial and agricultural by-products, specifically, run-of-mine perlite or rice husk ash, respectively.

Four LWC mixtures were optimized with respect to the achieved density and strength, and were evaluated in terms of durability. The main conclusions are presented in the following:

- The workability of lightweight concrete is influenced by both the physical characteristics of the alternative aggregates used and the pre-wetting procedure. Specifically, the combination of perlite fines with coarse pumice favoured the concrete's workability. The water trapped in the LWA pores during the pre-wetting, together with the sealing of the larger pores of the LWA by the slurry of Pe or Pu, favour and also maintain workability.
- All LWC mixtures, despite their ranking in low density classes (D1.6–1.8), developed an appropriate strength for structural use. Particularly, the incorporation of pumice and run-of-mine fine perlite yielded higher compressive strengths. Reaching a significant structural efficiency is vital, as it enhances and promotes the viability of the designed green concrete mixtures. A development of higher strength over time was also witnessed in all LWC mixtures containing Pe and Pu, relative to their content in the aggregate blend, as a result of their pozzolanic potential.
- Both sorptivity and open porosity values of the LWC mixtures increased, due to the porous nature of the used lightweight aggregates. However, the sorptivity of the mixture of perlite, coarse aggregate and pumice did not worsen significantly with respect to the REF, due to the relatively higher density of the mixture and the physical action of the fine grains of perlite. It seems that fine particles of perlite acted as filling agents and densified the microstructure and the interfacial zone of the concrete. An improvement in open porosity over time was also observed in the LWC mixtures that contained pumice, in agreement with their increase in strength.
- An excellent performance in terms of the resistance against chloride penetration was reported in all the LWC mixtures, which exhibited a further improvement with additional curing independent of the aggregate type. This fact can be accredited to the extended hydration and pozzolanic reactions, the binding capacity and the chloride pore-blocking effect. Moreover, all LWC mixtures at 90 days were classified correspondingly by both employed methods [37,39] with respect to their resistance against chloride penetration.
- In comparison to a conventional concrete mixture (REF), the LWC mixtures were susceptive to the FT cycles. The mixture of rice husk ash, perlite and pumice had the least favourable response after a few FT cycles, in relation to all the used FT indicators. The LWC mixtures with pumice also exhibited a low resistance against FT, although the combination of Pu with Pe exhibited a promising behaviour. Additional research, however, on the FT resistance of the optimized LWC mixtures is necessary.

From the experimental investigation carried out in this work, it was found that the examined LWC mixtures developed adequate strength and durability for structural use, with a remarkable resistance to chloride penetration. The optimized combination of pumice with perlite residues appears to be a sustainable proposal for LWC production and use, as it contributes to the preservation of natural resources whilst satisfying structural performance. This new type of LWC is in accordance with current regulations, and promotes recycling in the construction industry.

**Author Contributions:** Conceptualization, E.G.B. and V.G.P.; Formal analysis, M.C.S. and G.-E.D.L.; Investigation, M.C.S. and G.-E.D.L.; Methodology, M.C.S., G.-E.D.L., E.G.B. and V.G.P.; Project administration, M.C.S., E.G.B. and V.G.P.; Resources, E.G.B. and V.G.P.; Supervision, E.G.B.; Validation, M.C.S. and E.G.B.; Visualization, M.C.S., E.G.B. and V.G.P.; Writing—Original Draft, M.C.S.; Writing—Review and Editing, E.G.B. All authors have read and agreed to the published version of the manuscript.

**Funding:** This research received no external funding.

**Informed Consent Statement:** Not applicable.

**Data Availability Statement:** Data is contained within the article.

**Acknowledgments:** The authors gratefully acknowledge HERACLES, Group of Companies, Hellenic Concrete Technology Center (HCTC) for providing materials and laboratory when needed and the IMERYS INDUSTRIAL MINERALS GREECE S.A and EV.GE. Pistiolas S.A. for providing perlite and rice husk ash, respectively.

**Conflicts of Interest:** The authors declare no conflict of interest.

## Nomenclature

| | |
|---|---|
| *C* | Cement |
| *Ca* | Calcareous limestone aggregates |
| $D_{nssm}$ | Chloride penetration coefficient |
| $f_c$ | Compressive strength |
| *FT* | Freeze-thaw |
| *LWA* | Lightweight aggregates |
| *LWC* | Lightweight concrete |
| *OP* | Open Porosity |
| *Pe* | Run-of-Mine Perlite |
| *PL* | Sika® Plastiment 20R |
| *prW* | Water pre-conditioning |
| *Pu* | Pumice |
| *R* | Rice Husk Ash |
| *RCMT* | Rapid Chloride Migration Test |
| *RCPT* | Rapid Chloride Penetration Test |
| *S* | Sorptivity |
| *SP* | Sika® Viscocrete® Ultra-420 |
| *UPV* | Ultrasonic pulse velocity |

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
