# Peer review of "Perlite and Rice Husk Ash Re-Use As Fine Aggregates in Lightweight Aggregate Structural Concrete—Durability Assessment"

_sustainability, doi:10.3390/su15054217_

Round 1
Reviewer 1 Report
The paper "Perlite and rice husk ash re-use as fine aggregates in light weight aggregate structural concrete - Durability assessment” evaluates some LWA regarding durability. The paper fits into the scope of the journal and it is well-written.
However, as a reviewer, I have received in the last years the same type of paper, in which one changes a bit of a mixture and do thousands of tests. Then, the next move is to change another small thing and another paper. There is no WHY. I kept reading descriptions of tables and graphics, comparisons of the same things, and so on. Again, how much into the field of LWA/LWC the paper aforementioned has improved: little to nothing. I would love to see the authors evaluating WHY things have happened. Then one could suggest new materials with similar properties naturally.
I am saying this also because some of the authors have a similar paper – “A durability of Structural lightweight concrete containing different types of natural or artificial lightweight aggregates”. Again, the paper put under review is well-written and the findings are well-explained. But it is more of the same.
Author Response
Dear reviewer. Please find enclosed the file "Response to Reviewers 1 Comments.pdf" which contains the response of the authors to your comments.
Thanks once again for your effort.

Reviewer 2 Report
The author's work is appreciated however, the following further explanations are necessary for the manuscript to improve.
1. In the abstract, there is a lack of necessary data support.
2. The expression of “literature [11]; [12]” and others need to be improved.
3. The chemical composition in Table 1 is not 100%, Why?
4. In the mix design, there is a lack of value basis. For example, the Ca(4-8) in PeCaPu is 55kg/m3. What is the basis?
5. The crushing index of R, Pu and Pe should be different, How to consider their comparative study in this paper?
Author Response
Dear reviewer. Please find enclosed the file "Response to Reviewers 2 Comments.pdf" which contains the response of the authors to your comments.
Thanks once again for your effort.
